# *Porphyromonas gingivalis* Gingipains Destroy the Vascular Barrier and Reduce CD99 and CD99L2 Expression To Regulate Transendothelial Migration

Zhaolei Zou,[a] Juan Fang,[a] Wanting Ma,[a] Junyi Guo,[a] Zhongyan Shan,[a] Da Ma,[a] Qiannan Hu,[a] Liling Wen,[a] Zhi Wang[a]

[a]Hospital of Stomatology, Guanghua School of Stomatology, Guangdong Provincial Key Laboratory of Stomatology, Sun Yat-Sen University, Guangzhou, Guangdong, People's Republic of China

Zhaolei Zou, Juan Fang, and Wanting Ma contributed equally to this work. Author order was determined by drawing straws.

**ABSTRACT** *Porphyromonas gingivalis* is an important periodontal pathogen that can cause vascular injury and invade local tissues through the blood circulation, and its ability to evade leukocyte killing is critical to its distal colonization and survival. Transendothelial migration (TEM) is a series of that enable leukocytes to squeeze through endothelial barriers and migrate into local tissues to perform immune functions. Several studies have shown that *P. gingivalis*-mediated endothelial damage initiates a series of proinflammatory signals that promote leukocyte adhesion. However, whether *P. gingivalis* is involved in TEM and thus influences immune cell recruitment remains unknown. In our study, we found that *P. gingivalis* gingipains could increase vascular permeability and promote *Escherichia coli* penetration by downregulating platelet/endothelial cell adhesion molecule 1 (PECAM-1) expression *in vitro*. Furthermore, we demonstrated that although *P. gingivalis* infection promoted monocyte adhesion, the TEM capacity of monocytes was substantially impaired, which might be due to the reduced CD99 and CD99L2 expression on gingipain-stimulated endothelial cells and leukocytes. Mechanistically, gingipains mediate CD99 and CD99L2 downregulation, possibly through the inhibition of the phosphoinositide 3-kinase (PI3K)/Akt pathway. In addition, our *in vivo* model confirmed the role of *P. gingivalis* in promoting vascular permeability and bacterial colonization in the liver, kidney, spleen, and lung and in downregulating PECAM-1, CD99, and CD99L2 expression in endothelial cells and leukocytes.

**IMPORTANCE** *P. gingivalis* is associated with a variety of systemic diseases and colonizes in distal locations in the body. Here, we found that *P. gingivalis* gingipains degrade PECAM-1 to promote bacterial penetration while simultaneously reducing leukocyte TEM capacity. A similar phenomenon was also observed in a mouse model. These findings established *P. gingivalis* gingipains as the key virulence factor in modulating the permeability of the vascular barrier and TEM processes, which may provide a new rationale for the distal colonization of *P. gingivalis* and its associated systemic diseases.

**KEYWORDS** *Porphyromonas gingivalis*, gingipains, PECAM-1, CD99, CD99L2, transendothelial migration

Address correspondence to Zhi Wang, wangzh75@mail.sysu.edu.cn.

The authors declare no conflict of interest.

**P**orphyromonas gingivalis, an oral Gram-negative anaerobic bacterium, is an important pathogenic factor in periodontitis and can cause chronic inflammation or loss of periodontal support tissue (1, 2). In humans, *P. gingivalis* has been found to be potentially associated with cardiovascular disease, Alzheimer's disease, oral cancer, rheumatoid arthritis, colorectal cancer, and other systemic diseases (3–5). The pathogenicity of *P. gingivalis* is associated with its virulence factors, including lipopolysaccharides (LPS), hemagglutinins, outer membrane vesicles (OMV), and gingipains (6–8). Currently, gingipains are considered prominent virulence factors and play an important role in the pathogenesis of periodontal

disease, endothelial injury, and many other diseases (3, 9, 10). Gingipains include lysine-specific proteases (Kgp) and arginine-specific proteases (RgpA and RgpB) that can be expressed on the outer surface of the bacterium or secreted to perform their functions (1, 11). They affect proinflammatory cytokines like interleukin-1$\beta$ (IL-1$\beta$), IL-6, and IL-8 to modulate leukocyte migration (12–14).

The current opinion is that *P. gingivalis* can invade endothelial cells through the blood circulation, causing endothelial damage, and subsequently colonize local tissues (15). However, the ability of *P. gingivalis* to evade the relentless attack of leukocytes is essential for colonization and survival in distal tissues (16). Transendothelial migration (TEM) of leukocytes is a series of interactions between leukocytes and blood vessels, including the processes of rolling, activation, adhesion, locomotion, and transmembrane processes (17), that enable immune cells (neutrophils, monocytes, T cells, etc.) to transmigrate across endothelial vessels from the circulating bloodstream and enter local tissues to perform their effector functions. Various endothelial-junction molecules, such as platelet/endothelial cell adhesion molecule (PECAM-1), ICAM-1, VCAM-1, members of the junctional adhesion molecule (JAM) family, CD99, and CD99L2 (18, 19), have been reported to be involved in TEM. CD99 is a protein expressed on the surface of most leukocytes and at endothelial cell junctions and is involved in TEM processes of human and mouse monocytes and neutrophils through homologous interactions (20, 21). CD99L2, a CD99-associated protein (32% amino acid identity) expressed at endothelial cell junctions and on neutrophils, T cells, and B cells, is the only protein identified to date that is associated with CD99 and is also shown to be essential for the leukocyte TEM process (20–22). Previous studies have shown that *P. gingivalis* can reduce PECAM-1 expression on endothelial cells through gingipains (9) and can promote monocyte adhesion to the endothelium (23). However, the effect of *P. gingivalis* on TEM has not been investigated. In addition, the effect of *P. gingivalis* gingipains, proteases capable of cleaving a variety of junctional adhesion molecules, on CD99 and CD99L2, key TEM molecules expressed by both leukocytes and endothelial cells, is not known.

In this study, we performed a series of *in vitro* experiments to monitor the TEM process by which monocytes interact with the blood vessel barrier following *P. gingivalis*-mediated endothelial injury. We found that *P. gingivalis* reduced the TEM capacity of monocytes, despite promoting monocyte adhesion. Mechanistically, gingipains specifically reduced the expression of CD99 and CD99L2 by inhibiting the PI3K/Akt pathway in endothelial cells and multiple immune cell types. Furthermore, we observed that *P. gingivalis* colonization reduced the expression of PECAM-1, CD99, and CD99L2 in various tissues *in vivo*. These findings established gingipains as key virulence factors in modulating the permeability of the vascular barrier and TEM processes, which may provide a new rationale for the distal colonization of *P. gingivalis* and its associated systemic diseases.

## RESULTS

***P. gingivalis* gingipains increase vascular permeability and *Escherichia coli* penetration *in vitro*.** Previous studies showed that *P. gingivalis* alters the permeability of the vascular endothelial barrier, but the exact mechanism is unclear (24). To better study the specific effect of *P. gingivalis* on endothelium, a multiplicity of infection (MOI) of 100 was used for subsequent experiments. We established an *in vitro* fluorescein isothiocyanate (FITC)-dextran leakage model to examine vascular endothelial permeability by coculturing human umbilical vein endothelial cells (HUVECs) with or without *P. gingivalis* or *P. gingivalis* supernatant after centrifugation (Fig. 1a). We found that, compared with the control group, both the *P. gingivalis* and *P. gingivalis* supernatant groups showed significant changes in permeability ($P < 0.001$) (Fig. 1b). *P. gingivalis* gingipains are prominent virulence factors and were previously reported as secreted proteases that can be present in the supernatant (25); therefore, we investigated the effects of *P. gingivalis* strain ATCC 33277 and the gingipain-deficient *P. gingivalis* strain KDP 136 on vascular permeability. Brain heart infusion medium (BHI) was used as control group. We found that, strain ATCC 33277 significantly increased vascular permeability compared with that in the BHI group after 20 min ($P < 0.01$), while strain KDP 136 did not significantly alter vascular permeability (Fig. 1c). The permeability of the ATCC 33277 group was significantly higher than that of the KDP

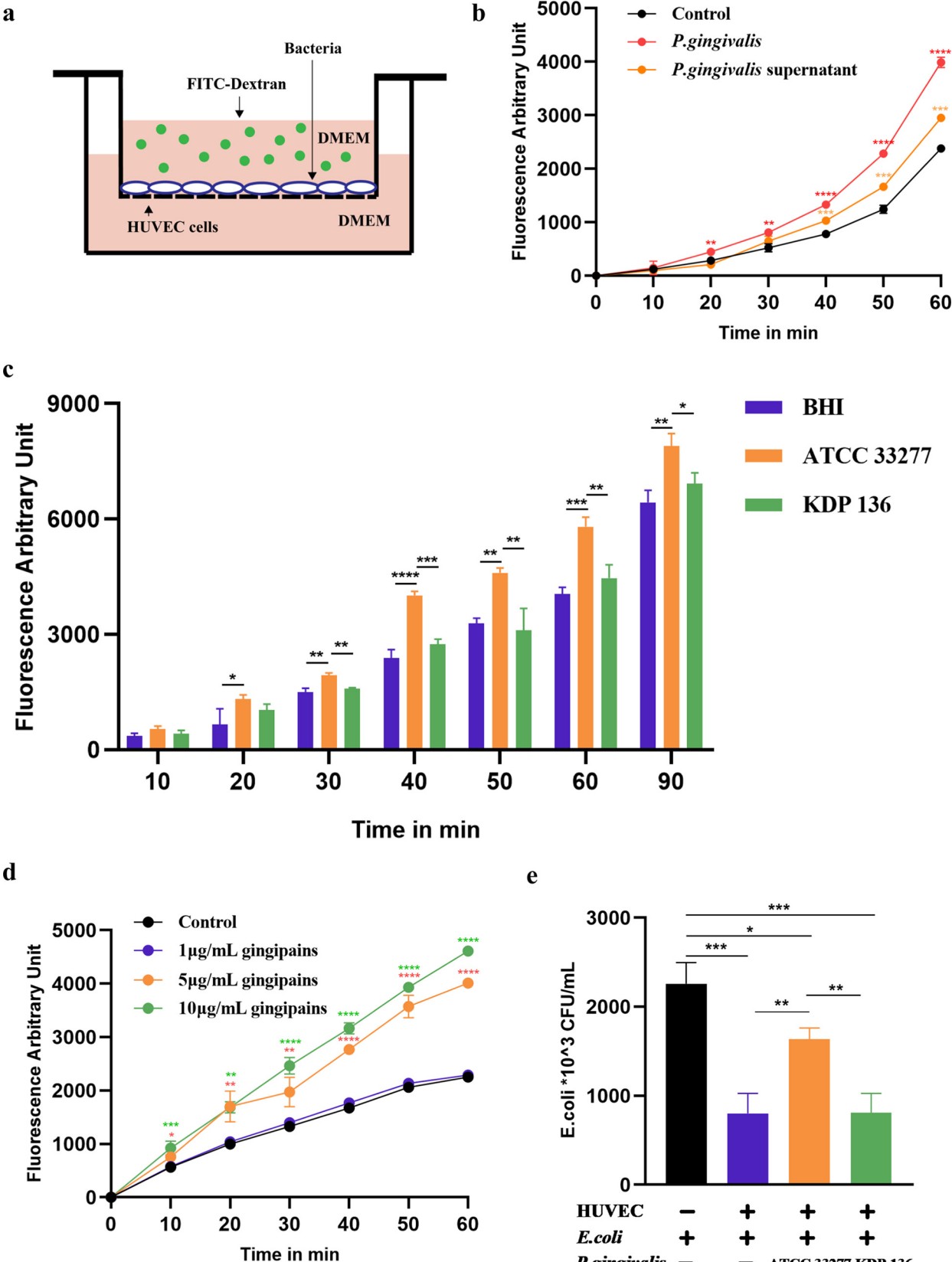

**FIG 1** *P. gingivalis* gingipains increase vascular endothelial permeability and promote *E. coli* penetration *in vitro*. (a) Schematic diagram of *in vitro* assay of vascular endothelial permeability. DMEM, Dulbecco modified Eagle medium. (b to d) Confluent HUVECs in transwells were cocultured

136 group after 30 min ($P < 0.01$) (Fig. 1c). Interestingly, the permeability change caused by gingipains was found to be dose dependent. The 1 $\mu$g/mL gingipains treatment showed no permeability change, while the 5 $\mu$g/mL and 10 $\mu$g/mL gingipains treatments significantly increased permeability after 10 min, with the greatest change observed with the 10 $\mu$g/mL treatment (Fig. 1d).

In addition, we determined the ability of *P. gingivalis* to facilitate the passage of other microorganisms, such as *E. coli*, through HUVECs. We found that the number of *E. coli* that penetrated was significantly reduced in the presence of HUVECs ($P < 0.001$), suggesting that the endothelial barrier blocked the penetration of *E. coli* (Fig. 1e). Compared with the control group and the KDP 136 group, the ATCC 33277 group showed significantly increased the penetration of *E. coli* (Fig. 1e). These results indicated that *P. gingivalis* gingipains increased vascular permeability and enhanced the penetration of *E. coli*.

**P. gingivalis gingipains reduce PECAM-1 protein expression.** Alterations in vascular permeability are associated with cytotoxicity or altered intercellular adhesion molecules (26). Cell counting kit-8 (CCK8) results showed that HUVECs infected with strain ATCC 33277 (MOI of 100) for 24 h had no change in cell viability, which indicated that gingipains did not cause permeability changes through cytotoxicity (Fig. S1 in the supplemental material). PECAM-1 and VE-cadherin have previously been shown to be critical for endothelial junctions to maintain endothelial barrier permeability (17). In our study, we found that both *P. gingivalis* and *P. gingivalis* supernatant decreased the protein levels of PECAM-1 and VE-cadherin on HUVECs compared with the levels in the BHI group (Fig. 2a). We also found that the protein levels of PECAM-1 and VE-cadherin were decreased after ATCC 33277 infection compared with the levels in the BHI group (Fig. 2b and c). However, the protein levels of PECAM-1 and VE-cadherin did not change significantly after strain KDP 136 infection. Interestingly, we found that PECAM-1 protein levels were nearly 1-fold lower after ATCC 33277 treatment than after KDP 136 treatment, while VE-cadherin protein levels were not significantly different (Fig. 2b and c). The real-time quantitative PCR results showed that ATCC 33277 and KDP 136 did not modulate PECAM-1 or VE-cadherin mRNA expression in HUVECs (Fig. 2d). Immunofluorescence (Fig. 2e) and flow cytometry (Fig. 2f) results confirmed that ATCC 33277 significantly reduced PECAM-1 expression on the HUVEC surface, while KDP 136 did not. Therefore, we speculated that gingipains reduced the protein expression level of PECAM-1 without affecting its mRNA expression level.

We next examined the effect of exogenous gingipains on PECAM-1 expression in HUVECs. We found that gingipains reduced the expression of PECAM-1 on HUVECs in a dose-dependent manner (Fig. 2g). Flow cytometry results also showed that gingipains reduced the expression of PECAM-1 on the HUVEC surface (Fig. 2h). These results suggested that *P. gingivalis* gingipains disrupt the endothelial barrier and increase vascular permeability *in vitro* by directly reducing PECAM-1 protein expression in HUVECs.

**P. gingivalis promotes THP-1 cell adhesion to HUVECs.** Following endothelial damage and dysfunction caused by *P. gingivalis*, the endothelium initiates a series of proinflammatory signals that promote leukocyte recruitment and adhesion (27). Here, we chose THP-1 cells as a tool to study leukocyte-vessel interactions. Previous studies have shown that *P. gingivalis* can promote the adhesion of THP-1 cells to HUVECs (28), and our results also found that the amount of adherent *P. gingivalis*-treated THP-1 cells increased 3-fold (Fig. 3a and b). Although *P. gingivalis*-treated HUVECs were able to adhere to more THP-1 cells, HUVECs adhered to fewer THP-1 cells of the ATCC 33277 group than of the KDP 136 group (Fig. 3c and d). Ligand/receptor pairs, such as ICAM-1/integrin $\alpha$M$\beta$2, are currently thought to be critical for monocyte adhesion to the endothelium (17). Western blotting (Fig. 3e) and flow

**FIG 1** Legend (Continued)

with *P. gingivalis* (MOI of 100) or culture supernatant from *P. gingivalis* strains ($1 \times 10^9$ CFU) (b), with strain ATCC 33277 or strain KDP 136 (MOI = 100) (c), or with increasing concentrations of gingipains (d). Endothelial permeability was assessed by adding FITC-dextran (1 mg/mL) to the upper chamber, aliquots were removed from the lower chamber, and fluorescence was measured at the indicated times. (e) *E. coli* (MOI of 100) was added to transwells with or without HUVECs or in combination with strain ATCC 33277 or strain KDP 136 (MOI of 100). After 1 h, aliquots were removed from the lower chamber and plated on LB agar plates, followed by incubation in air. Data are the mean values $\pm$ standard deviations (SD) from at least three independent experiments. *, $P < 0.05$; **, $P < 0.01$; ***, $P < 0.001$; ****, $P < 0.0001$. $P$ values are based on the $t$ test comparing the results to the control group (b, d) or on analysis of variance (ANOVA) followed by Tukey's test (c, e). MOI, Multiplicity of infection.

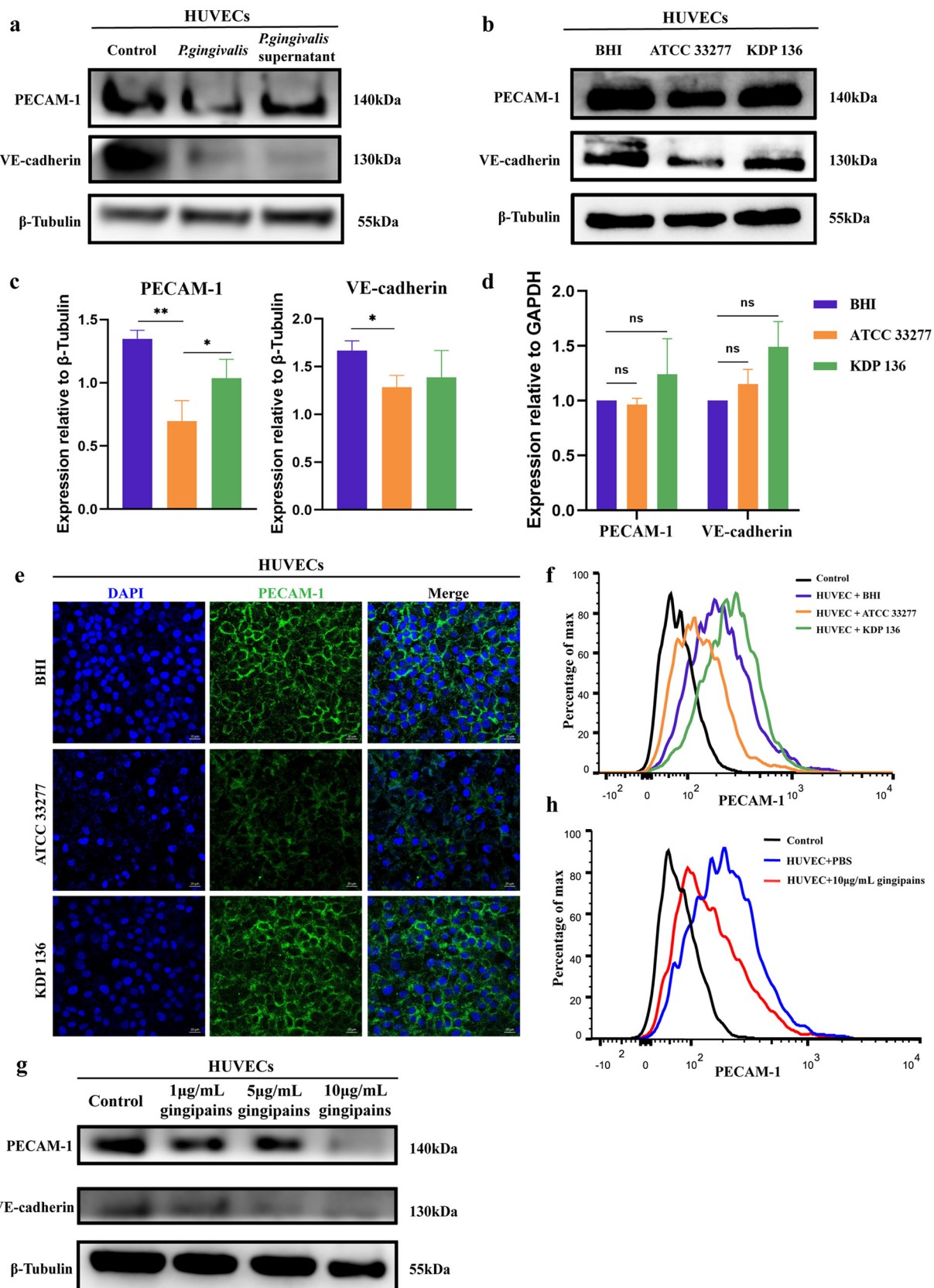

**FIG 2** *P. gingivalis* gingipains reduce PECAM-1 protein expression on HUVECs. (a) Western blots were used to examine the expression of PECAM-1 and VE-cadherin on HUVECs cocultured with phosphate-buffered saline (PBS), *P. gingivalis* (MOI of 100), or supernatant from *P. gingivalis* ($1 \times 10^9$ CFU) for 24 h. (b to f) HUVECs were cocultured with BHI, strain ATCC 33277, or strain KDP 136 (MOI of 100) for 24 h, and the expression of PECAM-1

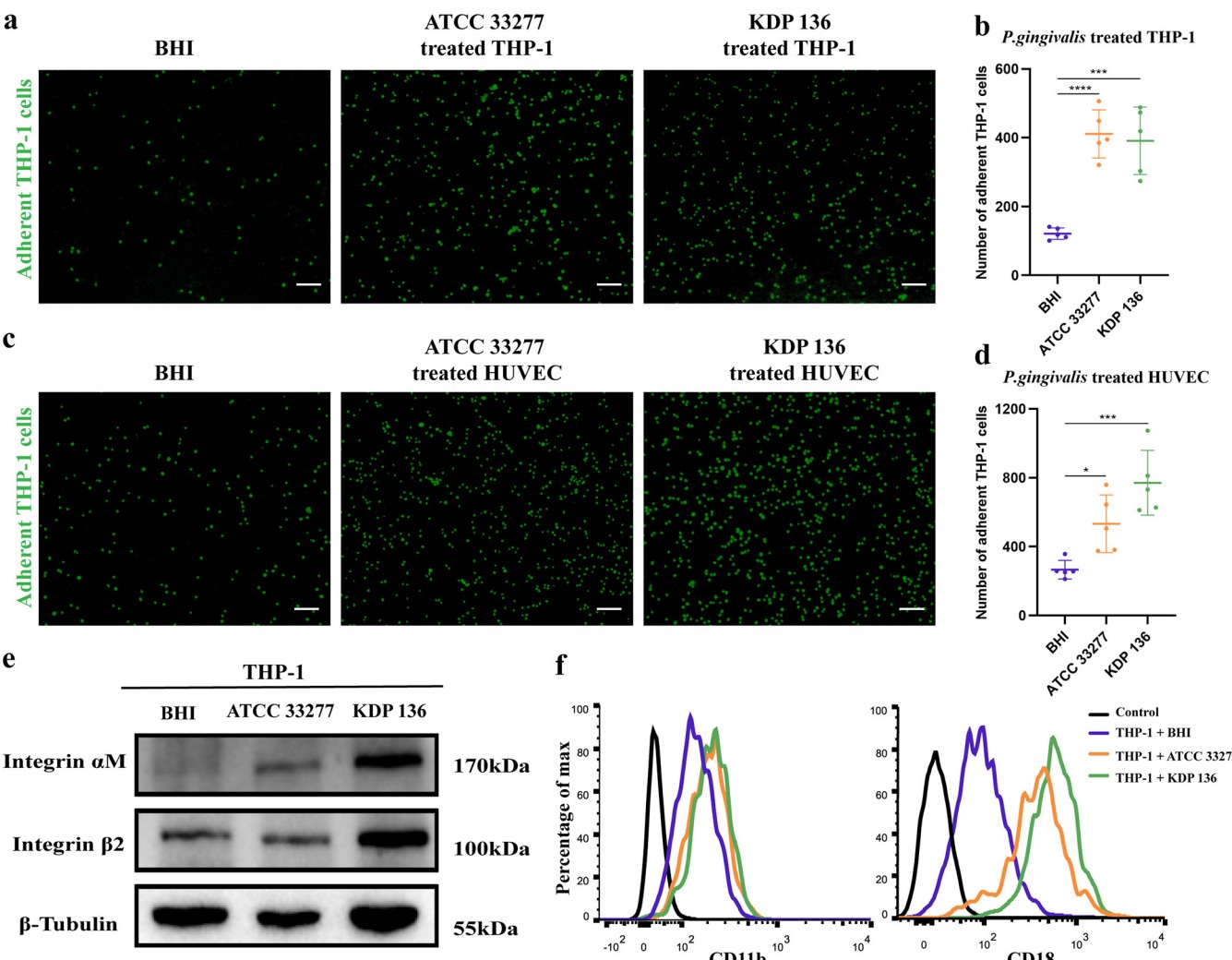

**FIG 3** *P. gingivalis* promotes THP-1 cell adhesion to HUVECs by facilitating integrin αMβ2 expression. (a, b) THP-1 cells were left untreated or cocultured with strain ATCC 33277 or strain KDP 136 for 24 h and labeled with calcein AM. Then, they were added to HUVECs cultured in 24-well plates and incubated for 2 h. The adherent THP-1 cells were imaged and counted by fluorescence microscopy. (c, d) HUVECs cultured in 24-well plates were left untreated or cocultured with ATCC 33277 or KDP 136 for 24 h, after which calcein AM-labeled THP-1 cells were added to the HUVECs and incubated for 2 h. The adherent THP-1 cells were imaged and counted by fluorescence microscopy. (e) Western blotting was performed to examine the expression of integrin αM (CD11b) and integrin β2 (CD18) by THP-1 cells infected with *P. gingivalis* for 24 h. (f) Flow cytometry was performed to examine CD11b and CD18 expression in THP-1 cells after treatment with BHI, ATCC 33277, or KDP 136 for 24 h. Data are the mean values ± SD from at least three independent experiments. *, $P < 0.05$; ***, $P < 0.001$; ****, $P < 0.0001$. (a, c) Scale bar = 100 $\mu$m.

cytometry results (Fig. 3f) showed that *P. gingivalis* enhanced the expression of integrin αM (CD11b) and integrin β2 (CD18) on THP-1 cells. Notably, we found that *P. gingivalis* gingipains reduced the expression of ICAM-1 in HUVECs (Fig. S2), which could explain the difference between the ATCC 33277 and KDP 136 groups shown by the results in Fig. 3d.

**_P. gingivalis_ gingipains reduce the TEM of THP-1 cells by degrading CD99 and CD99L2 expression on HUVECs.** TEM is the process by which leukocytes are recruited to function in tissues through a series of events, of which adhesion is the initiating phase (29). To investigate how *P. gingivalis* affects TEM, we established a monocyte chemoattractant protein-1 (MCP-1)-activated THP-1 cell TEM model *in vitro* using a transwell system. Compared with the BHI group and the strain KDP 136 group, the proportion of THP-1 cells that showed

**FIG 2** Legend (Continued)
and VE-cadherin was examined by Western blotting (b, c) and RT-qPCR (d). (e, f) The expression of PECAM-1 between HUVEC junctions was detected by Alexa Fluor 488-conjugated goat anti-rabbit secondary antibody using immunofluorescence microscopy (e) and flow cytometry (f). (g) HUVECs were incubated with the indicated concentrations of gingipains for 24 h, and the expression of PECAM-1 and VE-cadherin was detected using Western blotting. (h) Flow cytometry was used to examine the effect of gingipains on PECAM-1 on HUVECs. Data are the mean values ± SD from at least three independent experiments. *, $P < 0.05$; **, $P < 0.01$; ns, not significance. (e) Scale bar = 20 $\mu$m. MOI, multiplicity of infection.

TEM was reduced in the strain ATCC 33277 group ($P < 0.05$) (Fig. 4a). CD99 and CD99L2 are present at the junctions of endothelial cells and on the leukocyte surface; they facilitate the process of TEM through homologous interactions between two cell types and have been shown to be essential molecules in the completion of transmigration (20, 30). Therefore, we examined the expression of CD99 and CD99L2 on HUVECs after *P. gingivalis* infection. Western blotting results showed that strain ATCC 33277 significantly decreased the protein expression of CD99 and CD99L2 compared with their expression in the BHI group and strain KDP 136 group (Fig. 4b and c). We also found that KDP 136 reduced CD99L2 expression compared to its expression in the BHI group. RT-qPCR results showed that compared with the BHI group, ATCC 33277 decreased the mRNA expression levels of CD99 and CD99L2 in HUVECs, while no significant difference was observed in the KDP 136 group (Fig. 4d). We also observed the expression of CD99 in HUVECs by immunofluorescence microscopy and found that CD99 at the junction of HUVECs was significantly reduced in the ATCC 33277 group but not in the KDP 136 group (Fig. 4e). Flow cytometry results also showed that CD99 expression in HUVECs infected with strain ATCC 33277 was significantly lower than the CD99 expression in the BHI group and the strain KDP 136 group (Fig. 4f).

We next examined the effect of gingipains on the expression of CD99 and CD99L2 on HUVECs. Western blot analysis showed that 1 $\mu$g/mL gingipains significantly reduced the expression of CD99 and CD99L2 (Fig. 4g). Flow cytometry showed that 10 $\mu$g/mL gingipains significantly reduced the expression of CD99 on the surface of HUVECs (Fig. 4h). These results indicate that *P. gingivalis* reduced the TEM capacity of monocytes *in vitro* and that this process depended on gingipains degrading CD99 and CD99L2 on the surface of HUVECs.

**_P. gingivalis_ gingipains reduce CD99 and CD99L2 expression in leukocytes.** As previously mentioned, CD99 and CD99L2 are also expressed in leukocytes and influence the process of TEM (21, 22). Therefore, we examined the effect of *P. gingivalis* on CD99 and CD99L2 expression on THP-1 cells. Western blotting showed that strain ATCC 33277 was able to significantly reduce the expression of CD99 and CD99L2 on THP-1 cells (Fig. 5a and b). Meanwhile, CD99L2 expression in the strain KDP 136 group was lower than that in the BHI group. Immunofluorescence and flow cytometry showed similar results; ATCC 33277 downregulated the expression of CD99 on the THP-1 cell surface (Fig. 5c and d). In addition, exogenous gingipains reduced the expression of CD99 and CD99L2 on THP-1 cells (Fig. 5e). To further explore whether this phenomenon exists in other leukocytes, we used Jurkat cells and found that CD99 and CD99L2 were significantly downregulated after ATCC 33277 infection, as shown by both Western blotting and flow cytometry (Fig. S3).

The above-described results indicated that *P. gingivalis* degraded CD99 and CD99L2 on the surface of leukocytes, thereby affecting their TEM ability, which may affect the entry of leukocytes into tissues to perform their immune functions.

**_P. gingivalis_ gingipains reduce Akt phosphorylation.** Activation of the phosphoinositide 3-kinase (PI3K)/Akt pathway is associated with bacterial infection and plays an important role in the TEM process of leukocytes (neutrophils, monocytes, T cells, etc.) (11, 31, 32). Akt effectors are more commonly activated downstream from PI3K activation; thus, Akt phosphorylation is often used as a surrogate readout for PI3K activation (31). Therefore, we examined the Akt phosphorylation level after *P. gingivalis* infection and found that it was decreased by half in strain ATCC 33277-infected HUVECs compared to the levels in the BHI and strain KDP 136 groups (Fig. 6a). Similarly, we found reduced phosphorylation levels of Akt in THP-1 cells 24 h after ATCC 33277 infection (Fig. 6b). In addition, we found that the Akt phosphorylation levels of HUVECs and THP-1 cells decreased by half after treatment with 10 $\mu$g/mL gingipains compared to the Akt phosphorylation level in the control group ($P < 0.01$) (Fig. 6c and d). Furthermore, Erk1/2 pathway protein expression in HUVECs and THP-1 cells was examined by Western blotting, and the results showed that *P. gingivalis* gingipains promoted higher phosphorylation levels of Erk1/2 after 24 h of infection (Fig. S4).

These results suggested that *P. gingivalis* gingipains may inhibit Akt phosphorylation, thereby ultimately affecting the leukocyte TEM process.

**_P. gingivalis_ reduces PECAM-1 to increase vascular permeability and promotes bacterial penetration _in vivo_.** To further verify the results of the *in vitro* experiments, we established a mouse tail vein injection model of *P. gingivalis* to explore the effect

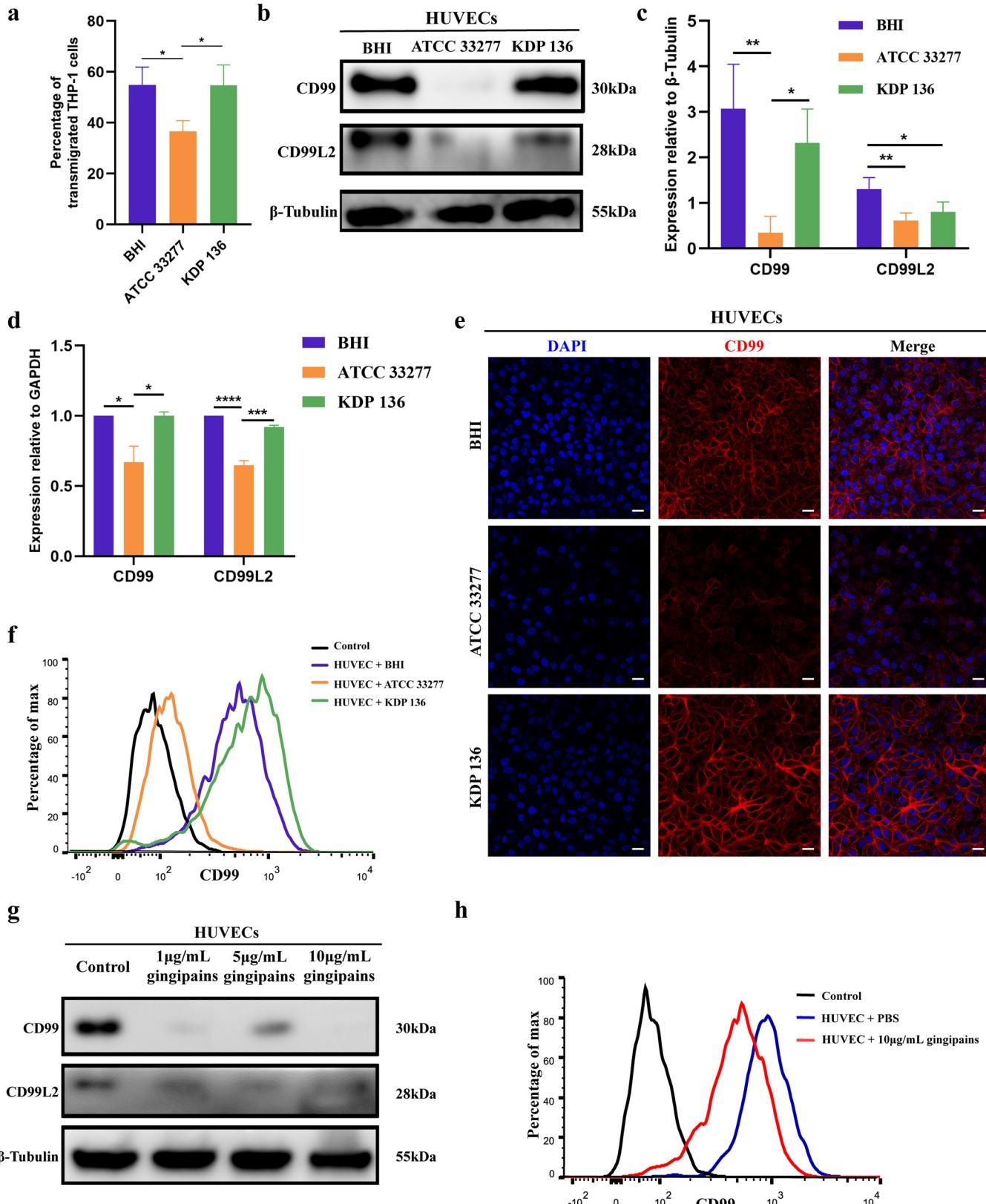

**FIG 4** *P. gingivalis* reduces the TEM of THP-1 cells by reducing CD99 and CD99L2 expression on HUVECs. (a) HUVECs were cultured with BHI, strain ATCC 33277, or strain KDP 136 in transwells for 24 h, and the MCP-1-activated-TEM assay was used to detect the TEM capacity of THP-1 cells. (b to f) HUVECs were cocultured with BHI, ATCC 33277, or KDP 136 for 24 h. (b, c) Western blotting was performed to examine the expression of CD99 and CD99L2 in *P. gingivalis*-infected HUVECs. (d) RT-qPCR was performed to examine the expression of CD99 and CD99L2 in *P. gingivalis*-infected HUVECs. (e) Immunocytochemistry images showing the expression of CD99 (red) in HUVEC junctions after *P. gingivalis* infection. (f) Flow cytometry results showing the expression of CD99 on HUVECs cocultured with BHI,

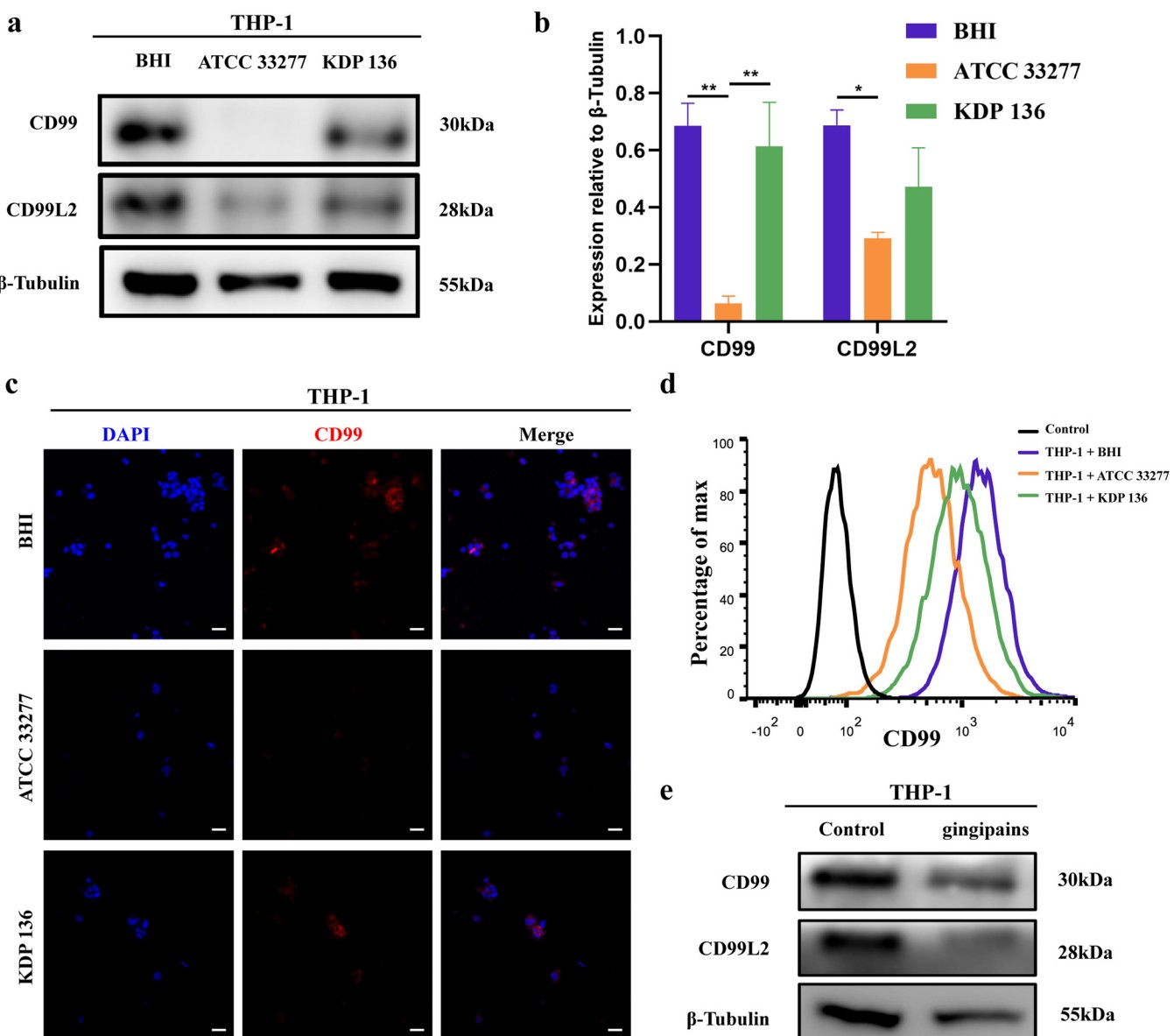

**FIG 5** *P. gingivalis* reduces CD99 and CD99L2 expression on THP-1 cells. THP-1 cells were cocultured with BHI, strain ATCC 33277, strain KDP 136, or gingipains for 24 h. (a, b) Western blotting to examine the expression of CD99 and CD99L2 in *P. gingivalis*-infected THP-1 cells. (c) Immunocytochemistry results showing the expression of CD99 (red) on THP-1 cells after *P. gingivalis* infection. (d) Flow cytometry results showing the expression of CD99 on the surface of THP-1 cells cocultured with BHI, ATCC 33277, or KDP 136. (e) Western blotting to examine the effect of 10 $\mu$g/mL gingipains on CD99 and CD99L2 expression on THP-1 cells. Data are the mean values $\pm$ SD from at least three independent experiments. *, $P < 0.05$; **, $P < 0.01$. (c) Scale bar = 50 $\mu$m.

on the vascular barrier *in vivo* (Fig. 7a). After 4 weeks, the vascular permeability of different organs of mice was detected by Evans blue assay. Compared with the sham group, the liver, kidney, spleen, and lung of the *P. gingivalis* group showed obvious permeability changes, while the heart, colon, pancreas, and brain did not show permeability changes (Fig. 7b). Furthermore, we detected the enrichment of total bacteria and *P. gingivalis* in the above-named permeability-altered tissues by RT-qPCR. We found that organs from the *P. gingivalis* group were more enriched in *P. gingivalis* than organs from the sham group, with the greatest enrichment in the liver (Fig. 7d). At the same time, the total bacterial numbers in the liver and kidney of the *P. gingivalis* group were also higher than those of the sham group (Fig. 7c).

**FIG 4** Legend (Continued)
ATCC 33277, or KDP 136. (g) HUVECs were incubated with the indicated concentrations of gingipains for 24 h, and Western blotting was performed to examine the expression of CD99 and CD99L2. (h) Results of flow cytometry examining the expression of CD99 in HUVECs after 10 $\mu$g/mL gingipains treatment. Data are the mean values $\pm$ SD from at least three independent experiments. *, $P < 0.05$; **, $P < 0.01$; ***, $P < 0.001$; ****, $P < 0.0001$. (e) Scale bar = 20 $\mu$m.

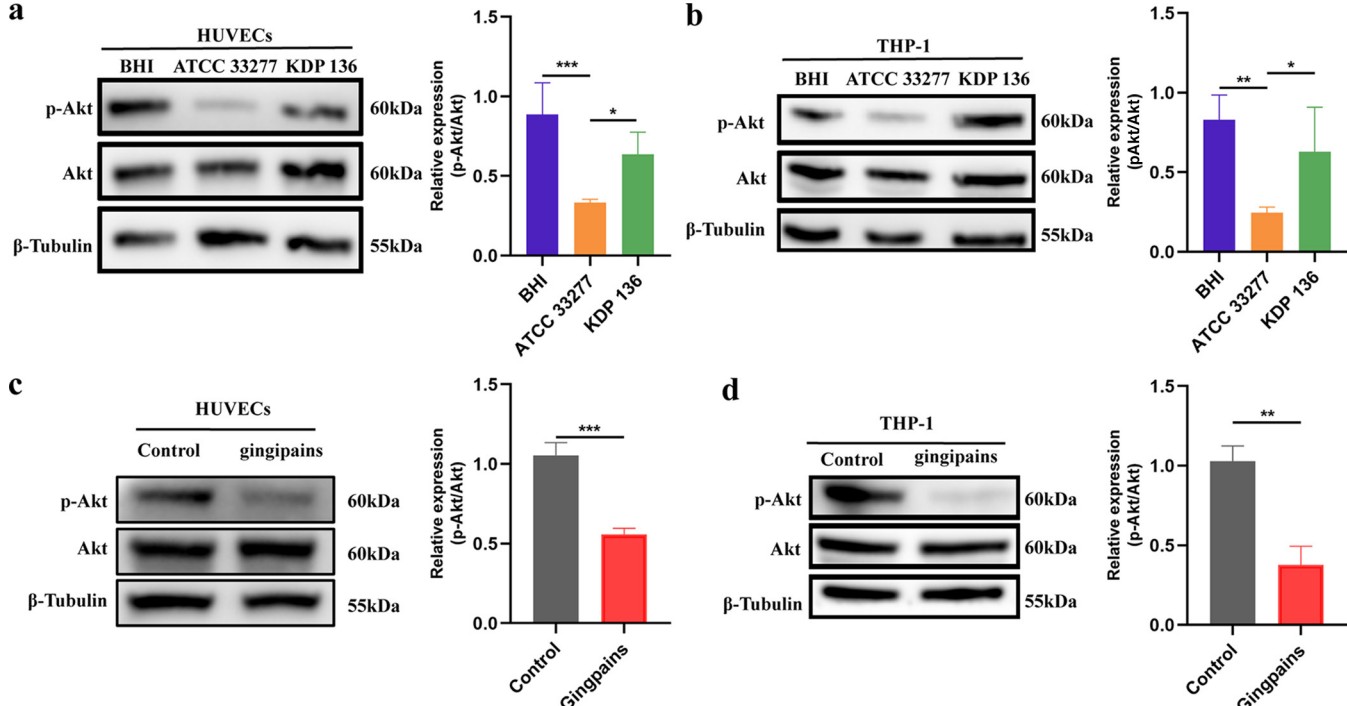

**FIG 6** *P. gingivalis* affects CD99 and CD99L2 expression by reducing Akt phosphorylation through a gingipain-dependent pathway. (a, b) Western blotting to examine the expression of p-Akt (S473) and Akt in HUVECs (a) and THP-1 cells (b) at 24 h after coculture with BHI, strain ATCC 33277, or strain KDP 136. (c, d) HUVECs (c) or THP-1 cells (d) were cocultured with 10 $\mu$g/mL gingipains, and Western blotting was performed to examine the expression of p-Akt (473) and Akt. Data are the mean values $\pm$ SD from at least three independent experiments. *, $P < 0.05$; **, $P < 0.01$; ***, $P < 0.001$.

Hematoxylin and eosin (H&E) staining results showed that, compared with the sham group, there were no obvious pathological changes or inflammatory infiltration in the organs of the *P. gingivalis* group (Fig. 7e).

Tissue immunofluorescence images revealed that PECAM-1 expression was downregulated in the blood vessels of the *P. gingivalis* group ($P < 0.0001$) (Fig. 7f and g). We next assessed the expression of PECAM-1 in various tissues of mice by immunohistochemistry (IHC) and found that the H-scores of PECAM-1 in the liver, kidney, spleen, and lung of the *P. gingivalis* group were significantly lower than those of the sham group (Fig. 7h and i). These results suggested that *P. gingivalis* enhanced vascular permeability and promoted microbial penetration by downregulating PECAM-1 *in vivo*.

***P. gingivalis* reduces CD99 and CD99L2 expression in mice.** We next examined whether tissue-colonizing *P. gingivalis* would affect CD99 and CD99L2 expression *in vivo*. IHC showed that the H-score of CD99 was lower in the *P. gingivalis* group than in the sham group in the liver, kidney, spleen, and lung (Fig. 8a and b). Western blotting of tissue homogenate proteins showed that the expression of CD99L2 in the kidney, spleen, and lung was significantly reduced after *P. gingivalis* injection (Fig. 8c and d). These results suggested that *P. gingivalis* can reduce tissue CD99 and CD99L2 expression *in vivo*, which may affect local leukocyte TEM capacity. This provides a new explanation for how tissue-colonizing *P. gingivalis* achieves immune escape (Fig. 8e).

## DISCUSSION

The mechanisms by which *P. gingivalis* colonizes local tissues and participates in systemic diseases have been the subject of much research in recent years (33). Much progress has been made in understanding the mechanism of leukocyte adhesion due to *P. gingivalis*-mediated endothelial injury (23, 34, 35). However, few studies have investigated the effect of *P. gingivalis* on the leukocyte TEM process. In our study, we aimed to illustrate the effect of *P. gingivalis* on leukocyte-vessel interactions and TEM processes. We found that *P. gingivalis* reduced PECAM-1 expression to increase vascular permeability

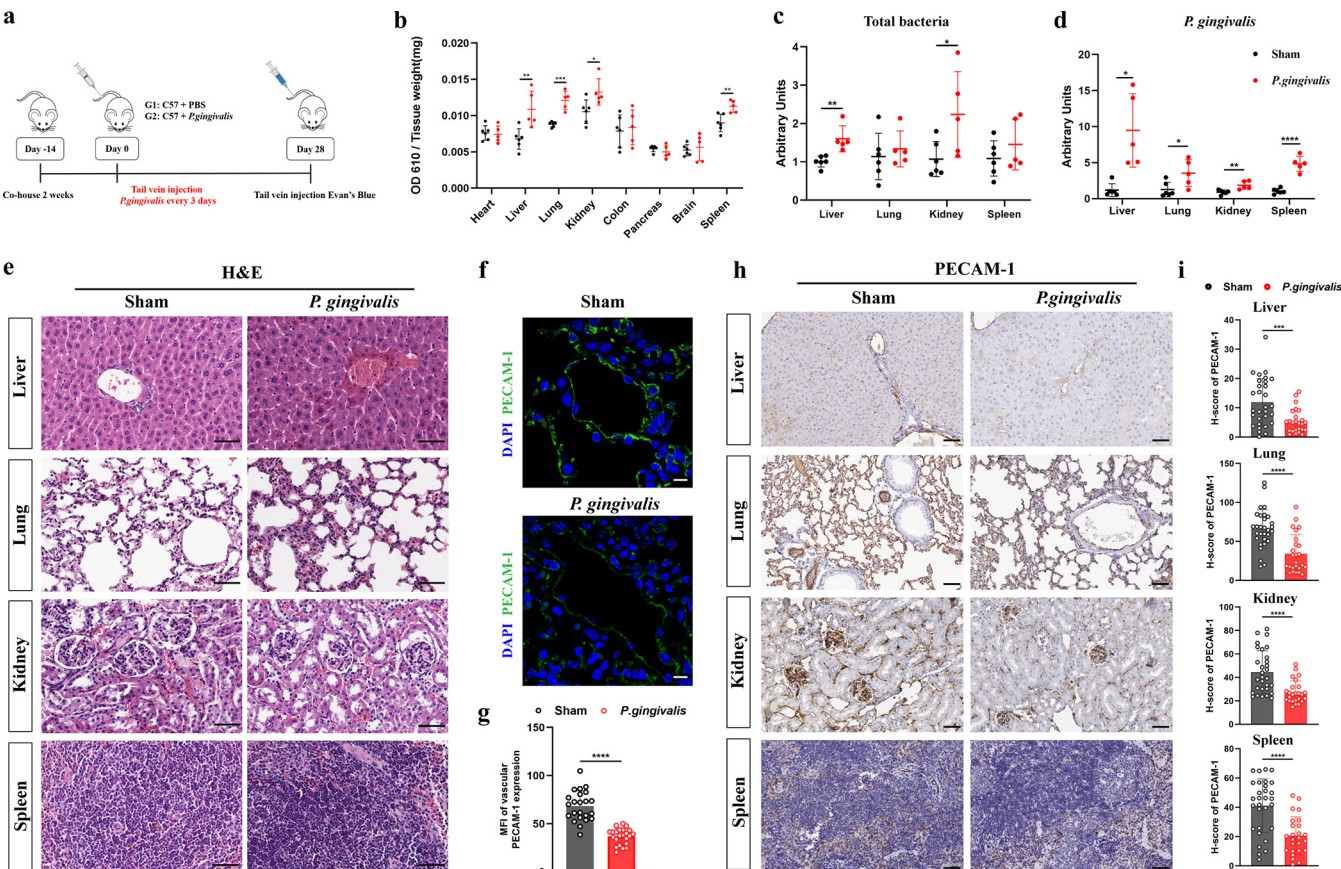

**FIG 7** *P. gingivalis* colonizes locally, disrupts the vascular barrier *in vivo*, and reduces PECAM-1 expression in blood vessels and tissues. (a) The sham group (*n* = 6) was tail vein injected with PBS, and the *P. gingivalis* group (*n* = 5) was injected with strain ATCC 33277 every 3 days for 4 weeks. (b) The Evans blue assay was used to detect vascular permeability in mice, and the penetration of the dye to the tissue was expressed by the ratio of optical density at 610 nm ($OD_{610}$) to tissue weight. (c) RT-qPCR was performed to detect the total bacteria enriched in different tissues. (d) RT-qPCR was performed to compare the enrichment of *P. gingivalis* in different tissues between two groups. (e) Representative images of hematoxylin and eosin (H&E) staining of liver, lung, kidney, and spleen tissue samples in the sham and *P. gingivalis* groups. (f) Representative immunofluorescence images showing the expression of PECAM-1 in mouse lung blood vessels. DAPI (blue), PECAM-1 (green). (g) Statistical graph representing the mean fluorescence intensities (MFI) of PECAM-1 in mouse lung blood vessels. (h) Representative immunohistochemistry (IHC) images showing the expression of PECAM-1 in liver, lung, kidney, and spleen tissue samples. (i) Statistical graphs representing the H-scores of PECAM-1 in liver, lung, kidney, and spleen tissues. Data are the mean values ± SD. *, $P < 0.05$; **, $P < 0.01$; ***, $P < 0.001$; ****, $P < 0.0001$. (e, h) Scale bar = 50 $\mu$m. (f) Scale bar = 20 $\mu$m.

and bacterial penetration. We also established *P. gingivalis* gingipains as the key factor in reducing CD99 and CD99L2 expression in endothelial cells and leukocytes to affect TEM capacity.

Previous studies have shown that *P. gingivalis* can reduce endothelial PECAM-1 expression to disrupt the endothelial barrier in periodontitis (8, 9). In our study, the reduction of PECAM-1 levels by *P. gingivalis* to increase vascular permeability was found to be dependent on gingipains (Fig. 1c and d and Fig. 2b to g). We demonstrated for the first time in a mouse model that *P. gingivalis* infection increased vascular permeability in the liver, kidney, spleen, and lung (Fig. 7b) by reducing vascular and tissue expression of PECAM-1 (Fig. 7f to i). The altered vascular permeability not only allows *P. gingivalis* to cross the endothelium but also allows other bacteria to pass through (36, 37). Thus, *P. gingivalis* acts as a facilitator for the systemic spread of other bacteria (38, 39). For the first time, we found that *P. gingivalis* can significantly increase the penetration capacity of *E. coli* (Fig. 1e). Accumulating evidence indicates that *P. gingivalis* is closely associated with renal dysfunction (40), nonalcoholic fatty liver disease (41, 42), and many other systemic diseases (4). Recently, *P. gingivalis* was found to trigger an imbalance in the Th17/Treg ratio in the liver and spleen (42). Our results showed that *P. gingivalis* was significantly enriched in liver, kidney, spleen, and lung tissues, with more bacteria in the liver and kidney *in vivo* (Fig. 7c and d). This provides new evidence for *P. gingivalis*-associated systemic disease in these enriched organs.

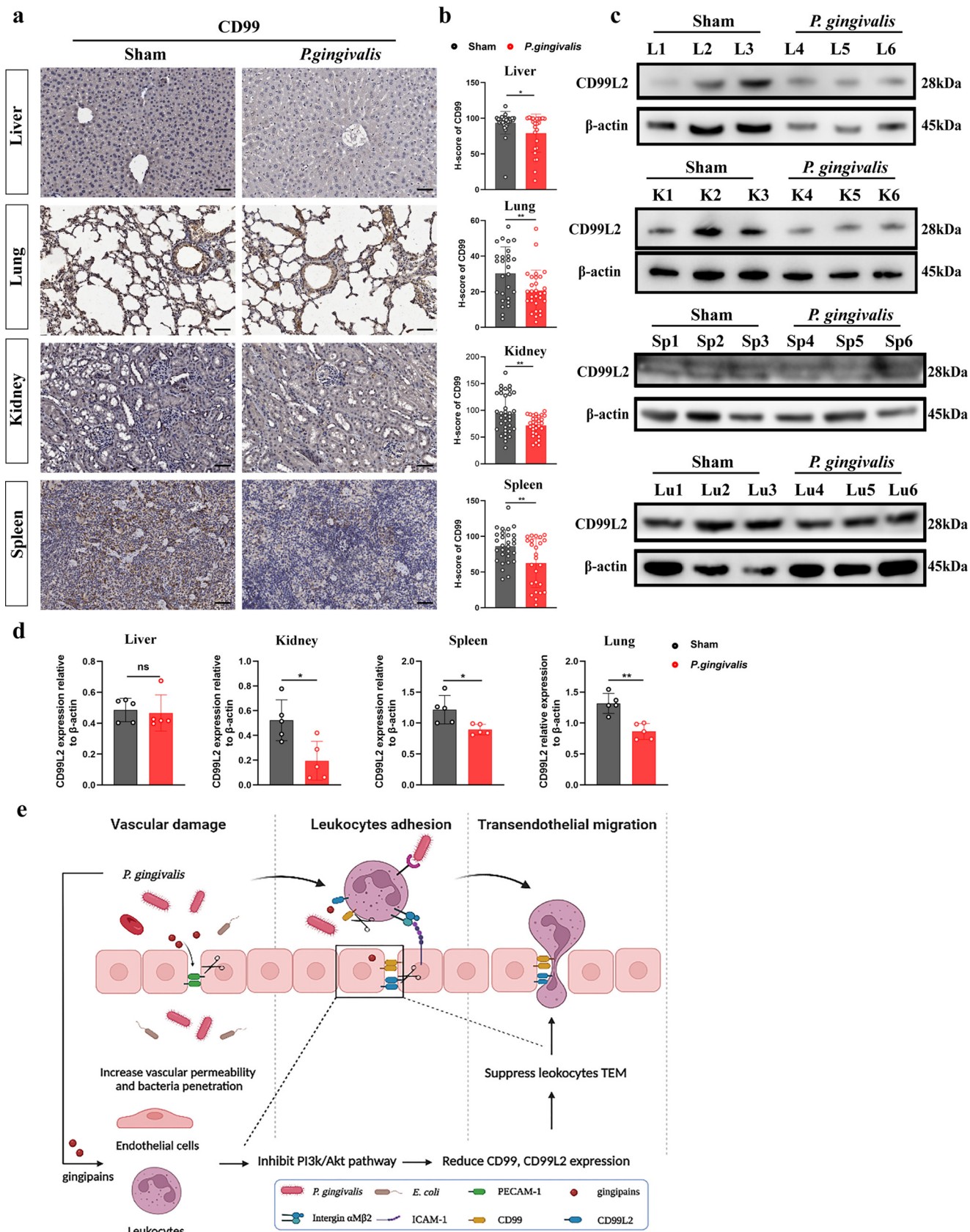

**FIG 8** *P. gingivalis* reduces tissue CD99 and CD99L2 expression. (a) Representative IHC images showing the expression of CD99 in mouse liver, lung, kidney, and spleen tissue samples. (b) Statistical graphs representing the H-scores of CD99 in mouse liver, lung, kidney, and spleen tissue samples. (c, d) Western blotting was

Leukocyte adhesion is the beginning of leukocyte-vessel interactions and TEM processes (43). Previous studies have shown that *P. gingivalis* fimbriae can promote monocyte adhesion to the endothelium by promoting ICAM-1/integrin $\alpha$M$\beta$2 binding (44, 45). Our results are similar in that *P. gingivalis* can stimulate the upregulation of THP-1 cell integrin $\alpha$M$\beta$2 expression to promote monocyte adhesion. Gingipains are reported to affect monocyte adhesion and recruitment in a concentration-dependent manner (46). Low concentrations of gingipains induce the expression of proinflammatory mediators like IL-1$\beta$, IL-8, IL-6, and ICAM-1 from epithelial cells, while at high concentrations, a reduction is observed (46). Interestingly, we found that THP-1 cell adhesion was lower in the *P. gingivalis* strain ATCC 33277 group than in the strain KDP 136 group, which we speculate is correlated with gingipains degrading ICAM-1 expression on HUVECs (47).

Yun and colleagues (48) found that gingipains reduced CD99 expression on HUVECs and suggested that it might affect leukocyte adhesion; however, the effect of gingipains on TEM has not been clarified. In the *in vitro* TEM assay, we were surprised to find that the number of transmigrated THP-1 cells was significantly lower in the strain ATCC 33277 group. This can be explained by the finding that *P. gingivalis* gingipains reduced the expression of both CD99 and CD99L2 on HUVECs, which are important functional molecules for leukocyte TEM and have been shown to be involved in the TEM of human and mouse neutrophils, monocytes, etc., independent of adhesion molecules like PECAM-1 (20). Blockade of CD99 or CD99L2 in mice by genetic knockout or specific antibodies resulted in the inhibition of the leukocyte TEM process (20, 30). These findings support our results showing that *P. gingivalis* gingipains reduce CD99 and CD99L2 expression on HUVECs to reduce the TEM capacity of leukocytes. More interestingly, for the first time, we observed a reduction in CD99 and CD99L2 on THP-1 and Jurkat cells after strain ATCC 33277 infection, and these results suggested that *P. gingivalis* gingipains can reduce CD99 and CD99L2 expression on both endothelial cells and multiple types of leukocytes, which in turn affects TEM capacity. In addition, we observed that the expression of CD99L2 on endothelial cells, THP-1 cells, and Jurkat cells was decreased significantly after strain KDP 136 treatment. *P. gingivalis* produces OMVs, which have been shown to affect the expression of a variety of adhesion molecules to perturb endothelial homeostasis (8). We postulated that in addition to gingipains, there were other factors in *P. gingivalis*, such as OMVs, that might affect CD99L2 expression.

The PI3K/Akt pathway is activated during T cell, monocyte, and neutrophil TEM processes and is also associated with bacterial infectious diseases (6, 32, 49). CD99-mediated TEM activates PI3K/Akt and increases Akt phosphorylation levels (50). In our study, *P. gingivalis* gingipains reduced CD99 and CD99L2 expression to affect TEM. Meanwhile, *P. gingivalis* gingipains reduced Akt phosphorylation levels in HUVECs and THP-1 cells. In Nakayama's study (11), gingipains were found to attenuate the PI3K/Akt pathway, leading to dysfunction of PI3K/Akt-dependent cells and destruction of epithelial barriers. This finding supports our results showing that *P. gingivalis* gingipains may affect the TEM process by attenuating the PI3K/Akt pathway.

Farrugia and colleagues (9) chose a zebrafish model of acute infection to study the mechanism of vascular damage caused by *P. gingivalis in vivo*. Since many of the current microbial-related systemic diseases involve prolonged exposure to microbial infection (51), we chose a mouse model with repeated inoculations with *P. gingivalis* to better mimic the mechanism *in vivo*. To meet animal welfare and ethical requirements, we tested the appropriate concentration of *P. gingivalis* injection. In the *P. gingivalis*-injected group, no significant inflammatory infiltration was found in the liver, kidney, spleen, or lung. At the same time, tissue PECAM-1, CD99, and CD99L2 expression levels were reduced, and these results suggest that *P. gingivalis* affected TEM capacity *in vivo* by reducing CD99 and CD99L2 expression and, thus, evaded immune killing by colonizing local tissues.

In conclusion, our study provides strong evidence that *P. gingivalis* gingipains are the

**FIG 8** Legend (Continued)
performed to examine the expression of CD99L2 in mouse liver, kidney, spleen, and lung tissue samples. (e) Schematic representation of the effects of *P. gingivalis* on leukocyte-vessel interactions and TEM processes. Data are the mean values $\pm$ SD. *, $P < 0.05$; **, $P < 0.01$; ns, no significance. L, liver; K, kidney; Sp, spleen; Lu, lung.

key virulence factor in the regulation of vascular permeability and TEM processes and can promote bacterial penetration. These effects promote the distal colonization of *P. gingivalis* and provide a rationale for *P. gingivalis* as a risk factor for certain systemic diseases.

## MATERIALS AND METHODS

All research materials and methods are listed in the supplemental material.  All animal study protocols were performed under the guidelines of the Institutional Animal Care and Use Committee of Sun Yat-Sen University (grant number SYSU-IACUC-2022-001542).

**Data availability.** We declare that all data supporting the findings of this study are available on reasonable request from the corresponding author.

## SUPPLEMENTAL MATERIAL

Supplemental material is available online only.
**SUPPLEMENTAL FILE 1**, PDF file, 0.6 MB.

## ACKNOWLEDGMENTS

We thank Jinlong Gao for a kind gift of gingipain-deficient *P. gingivalis* KDP 136.

We declare no conflict of interest.

Z.W. and Z.Z. conceptualized the study. Z.Z., J.F., and J.G. acquired most data and wrote the draft manuscript. Z.S., D.M., L.W., and W.M. offered help with data acquisition and statistical analysis. W.M. and J.F. performed critical revision on the manuscript. J.F. and Z.W. participated in the critical revision of the manuscript. Z.W. and J.F. were involved in funding acquisition. All authors approved the final version before submission.

This study was funded by the National Natural Science Foundation of China (grants number U22A20315, 81972532, and 82101017), the National Key Research and Development Program of China (grant number 2022YFC2402900), the Fundamental Research Funds for the Central Universities (Sun Yat-sen University) (grant number 87000-31670001), and the National Natural Science Foundation of Guangdong Province (grant number 2022A1515010771).

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
