## [Reviewer comments · Microbiology Spectrum]

Microbiology Spectrum

Porphyromonas gingivalis gingipains destroy the vascular barrier and reduce CD99 and CD99L2 expression to regulate transendothelial migration

Zhaolei Zou, Juan Fang, Wanting Ma, Junyi Guo, Zhongyan Shan, Da Ma, Qiannan Hu, Liling Wen, and Zhi Wang

Corresponding Author(s): Zhi Wang, Sun Yat-Sen University

Review Timeline:

Submission Date:	November 21, 2022
Editorial Decision:	December 16, 2022
Revision Received:	March 2, 2023
Accepted:	April 18, 2023

Editor: Jacqueline Abranches

Reviewer(s): The reviewers have opted to remain anonymous.

Transaction Report:

DOI: <https://doi.org/10.1128/spectrum.04769-22>

December 16, 2022

Prof. Zhi Wang
Sun Yat-Sen University
Sun Yat-Sen University, No. 56, Lingyuanwest Road, Guangzhou, Guangdong, P. R. China.
guangzhou 510055
China

Re: Spectrum04769-22 (Porphyromonas gingivalis gingipains destroy the vascular barrier and reduce CD99 and CD99L2 expression to regulate transendothelial migration)

Dear Prof. Zhi Wang:

Link Not Available

Sincerely,

Jacqueline Abranches

Journals Department
Reviewer comments:

Reviewer #1 (Comments for the Author):

Thank you for sending the revised manuscript of Zou et al. Authors made some efforts to improve the manuscript. Still, unfortunately, reviewer must admit that one critical issue was not resolved.

1) Figure 1e and supplementary Figure S1

I'm still not convinced using MOI=1000 in this manuscript. In rebutting letter, authors described that we experimentally confirmed the appropriate MOI. What does this "confirmed the appropriate MOI" refer to? Is that Supplementary Figure S1? Unfortunately, MOI and supernatants concentration did not show in Figure 1b. Thus, reviewer recommend that MOI and the concentration of

supernatant should be optimized in Figure 1b data. Additionally, the reason for selection (MOI=100) should be described in results section. Moreover, Figure 1e should be re-examined under the condition of using MOI=100, or MOI=10, 100, 1000. If MOI and the concentration of supernatant optimize in Figure 1b data, supplementary Figure S1 and S2 may be removed.

2) Line 156: "24 hours" should be revised to "24 h".

Reviewer #2 (Comments for the Author):

Lines 42-43: in which tissue/cells are PECAM-1, CD99 and CD99L2 expression downregulated? Was it in endothelial cells? The authors should include this detail in the abstract.

A classic mechanism of action of gingipains from *Porphyromonas gingivalis* is that this molecule cleaves human IL-8, thus modulating the number of neutrophils migrating to the site of infection (see references below). Therefore, gingipains may reduce the migration of inflammatory cells to the site of infection, while the authors are showing that gingipains increase the vascular barrier for leukocyte migration. The authors should take this piece of information into consideration in their introduction, data analysis, interpretations and discussion of their data. This should be added to the manuscript.

- Stathopoulou PG, Benakanakere MR, Galicia JC, Kinane DF. The host cytokine response to *Porphyromonas gingivalis* is modified by gingipains. *Oral Microbiol Immunol*. 2009 Feb;24(1):11-7. doi: 10.1111/j.1399-302X.2008.00467.x. PMID: 19121064; PMCID: PMC2717190.

- Mikolajczyk-Pawlinska J, Travis J, Potempa J. Modulation of interleukin-8 activity by gingipains from *Porphyromonas gingivalis*: implications for pathogenicity of periodontal disease. *FEBS Lett*. 1998 Dec 4;440(3):282-6. doi: 10.1016/s0014-5793(98)01461-6. PMID: 9872387.

- Sochalska M, Potempa J. Manipulation of Neutrophils by *Porphyromonas gingivalis* in the Development of Periodontitis. *Front Cell Infect Microbiol*. 2017 May 23;7:197. doi: 10.3389/fcimb.2017.00197. PMID: 28589098; PMCID: PMC5440471.

To date, are there any description in the literature on the role of gingipains from *Porphyromonas gingivalis* on the molecules that promote monocyte recruitment? This should be discussed in the "Discussion" section.

Figure legend 1: What is the rationale for using *P. gingivalis* at MOI of 100 in Fig 1c, and for using MOI of 1000 in Fig 1e?

Figure 1e: Why did the authors use MOI of 100 for *E.coli*, but MOI of 1000 for *P. gingivalis*? This rationale needs to be clearly stated in the manuscript.

Staff Comments:

Preparing Revision Guidelines

Please return the manuscript within 60 days; if you cannot complete the modification within this time period, please contact me. If you do not wish to modify the manuscript and prefer to submit it to another journal, please notify me of your decision immediately so that the manuscript may be formally withdrawn from consideration by Microbiology Spectrum.

Thank you for sending the revised manuscript of Zou et al. Authors made some efforts to improve the manuscript. Still, unfortunately, reviewer must admit that one critical issue was not resolved.

1) Figure 1e and supplementary Figure S1

I'm still not convinced using MOI=1000 in this manuscript. In rebutting letter, authors described that we experimentally confirmed the appropriate MOI. What does this "confirmed the appropriate MOI" refer to? Is that Supplementary Figure S1? Unfortunately, MOI and supernatants concentration did not show in Figure 1b. Thus, reviewer recommend that MOI and the concentration of supernatant should be optimized in Figure 1b data. Additionally, the reason for selection (MOI=100) should be described in results section. Moreover, Figure 1e should be re-examined under the condition of using MOI=100, or MOI=10, 100, 1000. If MOI and the concentration of supernatant optimize in Figure 1b data, supplementary Figure S1 and S2 may be removed.

2) Line 156: "24 hours" should be revised to "24 h".

Response to reviewer comments

Dear editor and reviewers,

Thanks very much for taking your time to review this manuscript. We appreciate all your comments and professional suggestions! Your comments are all valuable and very helpful for revising and improving our paper, as well as the important guiding significance to our researches. Based on your suggestion and request, we have made corrected modifications on the revised manuscript. We hope that our work can be improved again. Furthermore, we would like to show the details as follows:

Reviewer #1:

1. Figure 1e and supplementary Figure S1

I'm still not convinced using MOI=1000 in this manuscript. In rebutting letter, authors described that we experimentally confirmed the appropriate MOI. What does this "confirmed the appropriate MOI" refer to? Is that Supplementary Figure S1? Unfortunately, MOI and supernatants concentration did not show in Figure 1b. Thus, reviewer recommend that MOI and the concentration of supernatant should be optimized in Figure 1b data. Additionally, the reason for selection (MOI=100) should be described in results section. Moreover, Figure 1e should be re-examined under the condition of using MOI=100, or MOI=10, 100, 1000.

Author's response: We truly appreciate these constructive suggestions.

Firstly, we have added the MOI and concentration in Figure 1b figure legend and materials/method section. The details as follows, line 406-408, "...(b-d) Confluent

HUVECs in transwells were co-cultured with *P. gingivalis* (MOI=100), culture supernatant from *P. gingivalis* strains (1×10^9 CFU) (b)...;”, “To prepare for *P. gingivalis* supernatant, *P. gingivalis* ATCC 33277 was grown in BHI medium. Culture supernatant was collected after centrifugation ($6000 \times g$, 4°C , 10 min) and used for following experiment.”.

Secondly, we have added the reason for selection (MOI=100) in results section. We truly agree that it is difficult to distinguish whether it is a specific response of the organism or a general defence of the epithelial cells when the MOI is too large. The details as follows, line 113-115, “To better study the specific effect of *P. gingivalis* on endothelium, multiplicity of infection (MOI) = 100 was used for subsequent experiments.”

Thirdly, we truly agree with the suggestion on Figure 1e. We have re-done this experiment using MOI = 100. Moreover, we have revised Fig. S1. Please find the new Fig. 1e and Fig. S1 below. We hope these modifications could address your concerns.

Former Fig. 1e

New Fig. 1e

Former Fig. S1

New Fig. S1

2. Line 156: "24 hours" should be revised to "24 h".

Author's response: Line 143, "hours" has revised to "h".

Once again, we would like to thank you for all your time involved and this great opportunity for us to improve the manuscript. We hope you will find this revised version satisfactory. Thank you again for your suggestions!

Reviewer #2:

1. Lines 42-43: in which tissue/cells are PECAM-1, CD99 and CD99L2 expression downregulated? Was it in endothelial cells? The authors should include this detail in the abstract.

Author's response: We truly appreciate this constructive suggestion. In our IHC results, we found a reduction of these molecules in liver, lung, kidney, spleen, which contain endothelial cells and leukocytes. We have added this detail in the abstract.

The details as follows, line 41-43, "In addition, our in vivo model confirmed the role of *P. gingivalis* in promoting vascular permeability and bacterial colonization in the

liver, kidney, spleen and lung and in downregulating PECAM-1, CD99 and CD99L2 expression in endothelial cells and leukocytes.”.

2. A classic mechanism of action of gingipains from *Porphyromonas gingivalis* is that this molecule cleaves human IL-8, thus modulating the number of neutrophils migrating to the site of infection (see references below). Therefore, gingipains may reduce the migration of inflammatory cells to the site of infection, while the authors are showing that gingipains increase the vascular barrier for leukocyte migration. The authors should take this piece of information into consideration in their introduction, data analysis, interpretations and discussion of their data. This should be added to the manuscript.

Author’s response: We are grateful for this constructive suggestion. We truly agree that gingipains cleave IL-6, IL-8, MCP-1 to modulate leukocyte (neutrophil, monocyte...) migration. In our study, we found that gingipains could destroy the vascular barrier which cause more monocyte to adhere. However, when we using the MCP-1-activated-THP-1 TEM model, we found the number of migrated cells was lower in ATCC 33277 group. We suspect that gingipains could cleave key TEM molecules besides affecting proinflammatory cytokines. We have added this important information in our manuscript according to your helpful suggestion. The details as follows, line 66-69, “Gingipains include lysine-specific proteases (Kgp) and arginine-specific proteases (Rgp A and Rgp B) that can be expressed on the outer surface of the bacterium or secreted to perform their functions. They affect

proinflammatory cytokines like IL-1 β , IL-6 and IL-8 to modulate leukocyte migration”, line 324-328, “Gingipains are reported to affect monocyte adhesion and recruitment in a concentration-dependent manner. Low concentrations of gingipains induce the expression of proinflammatory mediators such as IL-1 β , IL-8, IL-6, and ICAM-1 from epithelial cells, while at high concentrations a reduction was observed.”. We have added these texts to address your concerns and hope it is clearer.

3. To date, are there any description in the literature on the role of gingipains from *Porphyromonas gingivalis* on the molecules that promote monocyte recruitment?
This should be discussed in the "Discussion" section.

Author’s response: We truly appreciate this suggestion. We have added in the discussion section. The details as follows, line 324-328, “Gingipains are reported to affect monocyte adhesion and recruitment in a concentration-dependent manner. Low concentrations of gingipains induce the expression of proinflammatory mediators such as IL-1 β , IL-8, IL-6, and ICAM-1 from epithelial cells, while at high concentrations a reduction was observed.”. Please find the reference below.

O'Brien-Simpson NM, Pathirana RD, Walker GD, Reynolds EC. *Porphyromonas gingivalis* RgpA-Kgp proteinase-adhesin complexes penetrate gingival tissue and induce proinflammatory cytokines or apoptosis in a concentration-dependent manner. *Infect Immun.* 2009;77(3):1246-1261. doi:10.1128/IAI.01038-08

4. Figure legend 1: What is the rationale for using *P. gingivalis* at MOI of 100 in Fig 1c, and for using MOI of 1000 in Fig 1e? Figure 1e: Why did the authors use MOI of 100 for *E.coli*, but MOI of 1000 for *P. gingivalis*? This rationale needs to be clearly stated in the manuscript.

Author’s response: We truly appreciate this constructive suggestion. We apologize for the confusion. We have re-experimented Fig. 1e using MOI = 100. Please find the new Fig. 1e below. We have revised the description in the figure legend and material/method section. The details as follows, line 412-413, “(e) ...with or without HUVECs, or in combination with ATCC 33277 or KDP 136 (MOI = 100).”

We hope you will find this revised version satisfactory. Thank you again!

Thank you very much for your time and attention! Look forward to hearing from you.

Yours sincerely,

Zhi Wang, Ph.D, DDS,

Professor of Oral Medicine,
Hospital of stomatology,
Guanghua School of Stomatology,
Guangdong Provincial Key Laboratory of
Stomatology, Sun Yat-Sen University,
No. 56, Lingyuanwest Road, Guangzhou, China,
510055 Tel: +86-20-87330592; +86-13829784088
Fax: 86-20-83822807
E-mail: wangzh75@mail.sysu.edu.cn

April 18, 2023

Prof. Zhi Wang
Sun Yat-Sen University
Sun Yat-Sen University, No. 56, Lingyuanwest Road, Guangzhou, Guangdong, P. R. China.
guangzhou 510055
China

Re: Spectrum04769-22R1 (Porphyromonas gingivalis gingipains destroy the vascular barrier and reduce CD99 and CD99L2 expression to regulate transendothelial migration)

Dear Prof. Zhi Wang:

Your manuscript has been accepted, and I am forwarding it to the ASM Journals Department for publication. You will be notified when your proofs are ready to be viewed.

Sincerely,

Jacqueline Abranches
Editor, Microbiology Spectrum

Journals Department
Issues have now been addressed to my satisfaction